# A Retrospective on eIF2A—and Not the Alpha Subunit of eIF2

**DOI:** 10.3390/ijms21062054

**Published:** 2020-03-17

**Authors:** Anton A. Komar, William C. Merrick

**Affiliations:** 1Center for Gene Regulation in Health and Disease, Department of Biological, Geological and Environmental Sciences, Cleveland State University, 2121 Euclid Avenue, Cleveland, OH 44115, USA; 2Department of Biochemistry, School of Medicine, Case Western Reserve University, Cleveland, OH 44106, USA; wcm2@case.edu

**Keywords:** eIF2A, translational initiation, alternative initiation, non-AUG initiation, stress response

## Abstract

Initiation of protein synthesis in eukaryotes is a complex process requiring more than 12 different initiation factors, comprising over 30 polypeptide chains. The functions of many of these factors have been established in great detail; however, the precise role of some of them and their mechanism of action is still not well understood. Eukaryotic initiation factor 2A (eIF2A) is a single chain 65 kDa protein that was initially believed to serve as the functional homologue of prokaryotic IF2, since eIF2A and IF2 catalyze biochemically similar reactions, i.e., they stimulate initiator Met-tRNA_i_ binding to the small ribosomal subunit. However, subsequent identification of a heterotrimeric 126 kDa factor, eIF2 (α,β,γ) showed that this factor, and not eIF2A, was primarily responsible for the binding of Met-tRNA_i_ to 40S subunit in eukaryotes. It was found however, that eIF2A can promote recruitment of Met-tRNA_i_ to 40S/mRNA complexes under conditions of inhibition of eIF2 activity (eIF2α-phosphorylation), or its absence. eIF2A does not function in major steps in the initiation process, but is suggested to act at some minor/alternative initiation events such as re-initiation, internal initiation, or non-AUG initiation, important for translational control of specific mRNAs. This review summarizes our current understanding of the eIF2A structure and function.

## 1. Introduction

Translation (protein synthesis) is the final step of expression of genetic information and is commonly separated into four phases: initiation, elongation, termination and ribosome recycling [1,2,3]. The initiation step is critically important, as it establishes the reading frame via the match between the AUG start codon in the mRNA and the unique, initiator transfer RNA (tRNA) aminoacylated with methionine. Elongation further governs the stepwise addition of the aminoacyl-tRNAs in a codon specific manner and the growth of the polypeptide chain. Termination occurs when the ribosome encounters a stop codon and leads to the release of the completed polypeptide chain. Ribosome recycling involves post-termination splitting of the ribosome into subunits followed by dissociation of deacylated tRNA and messenger RNA (mRNA) [1,2,3].

While the elongation step is highly similar between prokaryotes and eukaryotes, and termination and ribosome recycling are somewhat similar, the initiation step is strikingly different between prokaryotes and eukaryotes [2,3,4] (Figure 1). In prokaryotes, the proper selection of the initiation (start) codon depends on direct interaction between the small (30S) ribosomal subunit and the mRNA aided by the three single subunit initiation factors IF1, IF2, and IF3. IF3 binds to the small (30S) ribosomal subunit and prevents it from association with the large (50S) ribosomal subunit. Binding of IF2 and IF1 to the 30S-IF3 complex further promotes recruitment of the initiator *N*-Formylmethionine (fMet) tRNA (fMet-tRNA_i_) [2]. The 30S subunit binds to the mRNA template at a purine-rich (Shine-Dalgarno) sequence upstream of the initiation AUG codon. This allows direct positioning of the AUG (start) codon at the P-site of the 30S subunit and formation of the 30S initiation complex (30S IC). Association of the 30S IC with the 50S ribosomal subunit triggers hydrolysis of IF2-bound GTP and the machinery then proceeds to the elongation phase of translation [2]. IF2 is the sole and essential prokaryotic initiation factor that interacts specifically with initiator fMet-tRNA_i_ and aids its proper positioning in the ribosomal P-site to initiate translation [2].

Initiation in eukaryotes is a much more complex process and requires more than 12 different eukaryotic initiation factors (eIFs) [3,4,5,6,7,8]. Many of these factors are composed of multiple polypeptide chains [3,4,5,6,7,8]. For the majority of eukaryotic cellular mRNAs, translation starts with the recruitment of initiator Met-tRNA_i_ to the 40S ribosomal subunit by eukaryotic-specific initiation factor eIF2 (consisting of 3 different subunits α, β and γ). In cooperation with initiation factors eIF3, eIF1/1A and eIF5, the eIF2•GTP•Met-tRNA_i_ ternary complex (TC), binds the 40S ribosomal subunit yielding the 43S preinitiation complex (PIC) (Figure 1, right panel). eIF3 is one of the most complex eukaryotic translation initiation factors, consisting of multiple subunits (a to m) [9,10]. Similarly to IF3, eIF3 blocks small (40S) and large (60S) ribosomal subunit binding post-termination. It also plays important roles in TC recruitment to 40S and mRNA loading onto 43S [9,10]. eIF3 also promotes 43S binding to the 5′-end of mRNA via a bridge with eIE4G to initiate further scanning. The 43S complex scans the 5′-untranslated region (UTR) of eukaryotic mRNA in search of the initiating AUG codon to form the 48S PIC. eIF1A enhances TC binding to 40S subunit and mediates efficient start codon selection in tandem with eIF1 [3,5]. eIF1 also discriminates against non-cognate start codons or start codons in poor context [3,5]. Following recognition of the start codon and eIF5-induced irreversible hydrolysis of eIF2-bound GTP, eIF5B promotes joining of the 40S and 60S subunits and the elongation process begins [3,5]. The small (40S) ribosomal subunit serves as a central hub for initiation factors (i.e., eIF3, eIF1/1A/5 and TC) as well as mRNA and tRNAs [3,4,5,6,7,8]. After translation termination and ribosome recycling, the 40S and 60S subunits are dissociated, stripped of mRNA, P-site deacylated tRNA and polypeptide release factor eRF1 [3,4,5,6,7,8]. The process of ribosome recycling (aided by the ATP-binding cassette (ABC) protein ABCE1, eIF3, eIF1 and eIF1A) leads to “regeneration” of 40S subunits that become competent for initiation [3,4,5,6,7,8]. It is also important to note that at the beginning of initiation process, the initiation factor eIF4G functions as a scaffolding protein that bridges the mRNA and the ribosome [3,4,5,6,7,8]. It binds eIF4E (a 5′-end m^7^G cap-binding protein) and eIF4A (an ATP-dependent RNA helicase that unwinds secondary structure in the mRNA 5’ UTR). Binding of the mRNA to the 43S PIC is achieved by the mRNA bound eIF4G interaction with the 40S-bound initiation factor eIF3. Poly(A)-binding protein (PABP) is believed to enforce a “closed loop” mRNA conformation via interaction with eIF4G [3,4,5,6,7,8].

Much of the early work on characterizing the eukaryotic initiation factors was derived from four model systems, wheat germ, the aquatic crustacean *Artemia salina*, rat liver and rabbit reticulocytes [11,12,13,14,15,16,17,18,19,20,21,22,23,24,25,26,27,28,29,30,31,32,33,34,35]. These studies were triggered by findings in the bacterial system and used a majority of assays developed for this system (such as e.g., the use of codon triplets and Millipore filters for ribosomal binding studies with aminoacyl-tRNAs, the synthetic in vitro reaction of poly(U)-directed polyphenylalanine synthesis and synthesis of the first dipeptide, methionyl-puromycin) [36,37,38,39,40,41,42]. The ability to apply such assays was a key to the early characterizations of eukaryotic translation factors including those for initiation and those for elongation [11,12,13,14,15,16,17,18,19,20,21,22,23,24,25,26,27,28,29,30,31,32,33,34,35,42].

Using crude salt ribosomal washes, the initial characterizations of initiation factors included the binding reaction of the 40S ribosomal subunits to Met-tRNA_i_ in the presence of the start (AUG) codon [32], similar to the assay used in part to break the genetic code [36,37]. With these assays as the basis for activity determination, there were multiple reports on the partial or complete purification of a several proteins (protein fractions) with binding of initiator tRNA properties similar to the bacterial IF2 [13,14,16,18,19,20,21,22,23,24,25,26,27,28,29,30,33,34,35]. In addition, while there were the beginnings of a standardized nomenclature [42,43] for the bacterial proteins required for protein synthesis, in the mid-1970s and 1980s there were a variety of names and molecular weights for the eukaryotic proteins that served this function (see below).

This review summarizes our current understanding of the structure and function of a single subunit eukaryotic initiation factor 2A (eIF2A), that was initially believed to serve as the functional homologue of prokaryotic IF2, since eIF2A and IF2 were shown to catalyze biochemically similar reactions, i.e., they stimulated initiator methionyl-tRNA (Met-tRNA_i_) binding to the small ribosomal subunit in response to AUG codon [22,24]. However, in assays that used natural (globin) mRNA in place of the trinucleotide AUG, eIF2A was not found to be active [22,24]. Globin mRNA, is a classical cap-dependent mRNA that was predominantly used as a template in the mid-1970′s and 1980′s in in vitro protein synthesis assays [11,15,17,27,32]. Subsequent identification of a heterotrimeric factor, eIF2(α,β,γ) showed that this factor, and not eIF2A, is primarily responsible for the binding of Met-tRNA_i_ to 40S subunit in eukaryotes [44,45,46,47].

With the discovery of eIF2, the research on eIF2A languished. However, further discovery of a variety of alternative pathways for eukaryotic initiation (that may not rely on eIF2 and/or the presence of the m^7^G cap structure at the 5′ end of eukaryotic mRNAs) [3,48,49,50,51,52,53,54,55,56,57,58] generated a renewed interest in eIF2A and the role it might play in these alternative initiation events. In addition, it appeared that several other factors, and, specifically, Ligatin/eIF2D [59,60] and the complex of the oncogene product, MCT-1, and density regulated protein (DENR) [59] can promote recruitment of Met-tRNA_i_ to some 40S/mRNA complexes under conditions of inhibition of eIF2 activity. Thus, it became clear that factors like eIF2A may affect the translation of only a subset of mRNAs and this may happen under conditions when eIF2 is rendered less active.

## 2. Eukaryotic Initiation Factor 2 (eIF2)

While eIF2A was the factor first described as capable of mediating the binding of Met-tRNA_i_ to the 40S ribosomal subunits in eukaryotes [22,24], subsequent identification of eIF2 has clearly shown that this factor is the key player in eukaryotic translation initiation [44,45,46,47]. It is generally believed that under normal conditions eIF2 participates in the initiation of almost all cytoplasmic mRNAs in eukaryotic cells [3,5,58]. We will briefly describe eIF2 first, so that the differences in structure and function of eIF2A and eIF2 can be better understood.

eIF2 is a multimeric 126 kD protein complex consisting of three dissimilar subunits termed α, β, and γ [61,62] (Figure 2). Biochemical, genetic and structural studies have revealed that the three subunits have distinct functions [61,62]. The α-subunit has a molecular weight of 36 kDa and contains the main target residue for eIF2 regulation (through phosphorylation), a serine at position 51 [63,64]. The β-subunit (38 kDa) contains binding sites for eIF2B (the guanine nucleotide exchange factor) and eIF5 (a GTPase-activating protein) [65,66]. The γ-subunit is the largest (52 kDa) subunit and binds the guanine nucleotides, GTP and GDP and Met-tRNA_i_ [61,62,63,64,65,66].

The primary role of eIF2 is to deliver Met-tRNA_i_ to the 40S ribosomal subunit [61,62]. eIF2 does so in a ternary (eIF2•GTP•Met-tRNA_i_) complex comprising eIF2, GTP and Met-tRNA_i_ [61,62]. The affinity of Met-tRNA_i_ for eIF2 is affected by guanine nucleotides. eIF2 interacts with Met-tRNA_i_ with high affinity only when bound to GTP [67,68]. Following the placement of 40S ribosomal subunit at the AUG start codon at the end of the scanning process, eIF5 stimulates the hydrolysis of eIF2-bound GTP [69,70]. The eIF2•GDP complex has a low affinity for Met-tRNA_i_ and leaves the ribosome together with eIF5 [69,70,71]. Prior to participation in the next rounds of initiation and binding of Met-tRNA_i_, the GDP in the eIF2•GDP complex has to be exchanged for GTP [66,72,73]. This reaction is catalyzed by eIF2B, a protein complex consisting of 5 subunits: α (26 kDa), β (39kDa), γ (58kDa), δ (67kDa) and ε (82kDa) [66,72,73]. Structural studies of eIF2B have shown that it is a decamer or a dimer of pentamers [66]. eIF2 activity in eukaryotes is highly regulated in response to several physiological stresses [3,5,6,7,8], which in particular include changes in intracellular accumulation of unfolded or denatured proteins, virus infection, heme deficiency and nutrient/amino acid deprivation [63,64].

In mammals, four stress-activated kinases (PKR-like ER kinase (PERK), double-stranded RNA-dependent protein kinase (PKR), heme-regulated eIF2α kinase (HRI), and general control non-derepressible 2 kinase (GCN2) reduce the level of active eIF2 by phosphorylating the eIF2α subunit and, consequently, reducing the global level of translation [63,64]. Phosphorylation of the α subunit of the eIF2 complex inhibits eIF2B-mediated exchange of eIF2•GDP to eIF2•GTP [66,72,73], thus reducing the formation of the active TC [63,64]. However, translation of a subset of cellular and viral mRNAs appeared to be resistant to eIF2α phosphorylation, despite requiring Met-tRNA_i_ [3,48,49,50,51,52,53,54,55,56,57,58]. These mRNAs in many instances do not require m^7^G cap recognition by eIF4E, and their translation may rely on alternative initiation mechanisms such as internal initiation and/or re-initiation [3,50,51,52,53]. It was found that a subset of factors, i.e., Ligatin/eIF2D [59,60], the oncogene MCT-1 and DENR (together) [59] as well as eIF5B (alone) [49,74,75] can promote efficient recruitment of Met-tRNA_i_ to 40S/mRNA complexes under conditions of inhibition of eIF2 activity, or its absence.

eIF2A was originally suggested to function in the same pathway, i.e., promote recruitment of Met-tRNA_i_ to 40S ribosomal subunits [22,24]. However, its precise role, as well as the precise mechanism of its action, still remain largely enigmatic as many reports describing eIF2A function seem to contradict one another.

## 3. eIF2A—Discovery and Initial Characterization

As has been mentioned above, the early work on characterizing the eukaryotic initiation factors [11,12,13,14,15,16,17,18,19,20,21,22,23,24,25,26,27,28,29,30,31,32,33,34,35] was guided by findings in the bacterial systems [38,39,40]. Following these studies researchers were aiming to find eukaryotic factors capable of binding Met-tRNA_i_ to 40S subunits in a codon-dependent manner. Unlike in bacteria, mRNA interaction with the 40S subunit in eukaryotes only occurs after Met-tRNA_i_ binding to the partial P-site. Because of this, it was not understood that eIF2 was capable of binding Met-tRNA_i_ to the 40S ribosomal subunit in a GTP-dependent, but AUG-independent manner [20,21,29,30,44,45,46]. Thus, the presence of GTP would give the “false” impression of a high background (i.e., not AUG-dependent).

However, it became apparent that formation of an elongation competent 80S initiation complex requires GTP, as well as AUG triplet and some additional factors [11,12,13,14,15,16,17,18,19,20,21,22,23,24,25,26,27,28,29,30,31,32,33,34,35]. The use of the poly(U)-directed synthesis of polyphenylalanine Mg^2+^ shift assay also revealed that the elongation factors eEF1A and eEF2, and some other proteins (later to be identified as eIF1A, eIF5A and eIF5B) are required for this process [22,47].

With these assays as the basis for activity determination, there have been multiple reports on the partial or complete purification of a variety of proteins of different molecular composition and weight capable of binding of initiator tRNA_i_ to the 40S ribosomal subunit in an AUG-dependent manner (similar to that of the bacterial IF2) (Table 1).

At the same time, multiple laboratories had identified a protein that bound Met-tRNA_i_ to 40S ribosomal subunits in a GTP-dependent manner [20,21,29,30,44,45,46], a protein that would become known as eIF2. The discovery of eIF2, coupled with the first regulation of protein synthesis by heme in the reticulocyte system [see 47 for a review], quickly led to decades of research on eIF2 and the regulation of translation through the phosphorylation of serine 51 in the α-chain of this three subunit protein [61,62]. Curiously, however, in the original model assay, AUG-directed methionyl-puromycin synthesis, it was shown that both eIF2 and IF-M1 (later became known as eIF2A) could participate in this reaction with exactly the same required co-factors (eIF1A, eIF5A and eIF5B) [24].

The original purification procedure that led to the identification of eIF2A (at that time called IF-M1) involved preparation of ribosomal salt wash from rabbit reticulocytes (step 1), ammonium sulfate precipitation (step 2), DEAE-chromatography (step 3), subsequent additional ammonium sulfate precipitation (step 4), Sephadex G-200 chromatography (step 5), CM-cellulose chromatography (step 6), phosphocellulose chromatography (step 7) and Sephadex G-100 chromatography (step 8) as key purification steps [22,24]. The above eight-step procedure yielded a homogeneous (as judged by SDS-gel electrophoresis) preparation of the 65 kDa protein, which was named IF-Ml [22]. The amino acid composition of IF-M1 was also determined after protein hydrolysis in methanesulfonic acid [22]. IF-M1 activity was further tested in several model assays mentioned above. These, in particular, included formation of puromycin-sensitive 80S initiation complexes with Met-tRNA_i_, poly(U)-directed polyphenylalanine synthesis, AUG and ribosome-dependent Met-tRNA_i_, binding and some other assays [22,24]. In these model assays, IF-M1 has been shown to bind Met-tRNA_i_ and deliver it to the ribosome in a GTP-independent but AUG-dependent manner [22,24]. IF-M1 was also shown to be able to catalyze poly(U)-directed polyphenylalanine synthesis in low [Mg^2+^], but unable to support protein synthesis with globin mRNA [22,24]. Because different results were obtained with different IF-M1 preparations, additional studies were performed to test the properties of IF-Ml activity during different stages of purification. From these studies, a non-lF-Ml Met-tRNA_i_ binding protein was also detected in one of the fractions between purification steps 6 and 7 that appeared to have a molecular weight similar to IF-M1, as well as quite similar, but not identical chromatographic characteristics (similar DEAE- and CM-cellulose, but not phosphocellulose binding) and similar Met-tRNA_i_ binding activity, requiring AUG template and ribosomes [22]. It cannot be excluded that this protein would be subsequently identified as ligatin/eIF2D [59,60]. These subsequent experiments, which led to the identification of eIF2D, employed a modified purification strategy, which nevertheless involved many similar steps (ammonium sulfate precipitation, DEAE-cellulose, phosphocellulose, MonoQ and MonoS chromatography) [59,60]. These experiments also yielded eIF2A [60]. eIF2A isolated in this later study did not appear to possess Met-tRNA_i_ binding activity [60], potentially suggesting that the protein might have been modified during the isolation, or missing a cofactor required for its activity.

More recent studies have identified other proteins like the MCT-1/DENR complex [59] and eIF5B [49,74,75], which also participate in binding of aminoacyl-tRNAs to ribosomes. The best evidence for the function of these non-eIF2 proteins is their ability to function in some of the model assays of protein synthesis which include binding of aminoacyl-tRNA to 40S ribosomal subunits in a codon-dependent manner, synthesis of aminoacyl-puromycin or the poly(U)-directed synthesis of polyphenylalanine and toe-printing (Table 2).

Of these four proteins, two (eIF2A and eIF2D) seemed to be highly similar in molecular weight and as has been mentioned above to be co-purified through multiple chromatographic steps [22,24,59,60]. Both proteins are about 65,000 Da and single polypeptide chains [22,24,59,60]. However, their pI and amino acid compositions appeared to be different as can be seen in Table 3. The identification of the cDNA of human eIF2A (GeneBank™ accession number AF497978) [76]) and its predicted amino acid sequence has unambiguously shown that IF-M1/eIF2A and eIF2D are two different proteins (Table 3).

## 4. Identification of the Mammalian eIF2A Gene and the Yeast Homologue

In-depth analysis of eIF2A was facilitated by the identification of the eIF2A gene and the development of a convenient model system allowing assessment of the eIF2A function and activity on the cellular level. These studies were accomplished only in 2002 [76], after more than 25 years, since the original paper describing isolation of IF-M1/eIF2A was published [22]. Using tryptic peptides from a preparation of the rabbit reticulocytes eIF2A, we were able to deduce partial amino acid sequence of the protein and subsequently identify the eIF2A gene [76], which, indeed, appeared to be encoding a 585 amino acid protein [76], as was originally suggested [22,24].

Furthermore, the amino acid composition of the rabbit eIF2A protein deduced from the rabbit eIF2A gene sequence appeared to be quite similar to that originally reported for IF-M1 [22] and dissimilar from that of eIF2D (Table 3; compare, e.g., the number of such characteristic amino acid residues as Leu, Lys, Phe and Trp between the proteins). Using the cDNA information, we were further able to map the position of the eIF2A gene in the human genome, which was found to be located on chromosome 3 with transcription heading away from the stress-associated endoplasmic reticulum protein 1 (SERP1) [76].

Identification of the mammalian eIF2A gene further allowed a search for a yeast *Saccharomyces cerevisiae* homologue (*YGR054W*), which was found to be 28% identical (58% similar) to human eIF2A [76]. Importantly, eIF2A protein homologues were found in a wide range of eukaryotic species (Figure 3), suggesting a conserved biological role.

## 5. eIF2A Function in Yeast *S. cerevisiae*

To begin the functional characterization of the yeast *S. cerevisiae* eIF2A, we at first obtained and characterized the Δ*eIF2A* yeast strain with a disrupted copy of the eIF2A gene (*YGR054W*) [76]. The Δ*eIF2A* strain appeared to be viable and had a doubling time of 1.3 h, nearly identical to that of the wild-type (1.2 h) [76,77]. The lack of any apparent phenotype in the Δ*eIF2A* strain and the lack of any visible polysome profile defect in this null mutant further suggested that eIF2A is not functioning in a major initiation pathway, but may function in a minor pathway, perhaps, internal initiation or in the translation of a small number of specific mRNAs [76,77].

Interestingly, however, the double deletion ΔΔ*eIF2A/eIF5B* and Δ*eIF2A/eIF4E-ts* mutant *S. cerevisiae* strains displayed a severe slow growth phenotype [76,77]. The phenotype of these mutants and the biochemical localization of the eIF2A on the 40S ribosomal subunits as well as 80S ribosomes further suggested that eIF2A participates in translation initiation [76,77]. The observed genetic interaction between eIF2A and eIF4E as well as eIF5B was taken further by Davey et al. [78] and Kim et al. [79], who were able to detect a direct physical interaction between eIF2A-eIF4E and eIF2A-eIF5B proteins, respectively, using pull down assays of tagged proteins (see below). Our analysis of eIF2A function in eukaryotic initiation in yeast cells further showed that eIF2A does not affect cap-dependent initiation or re-initiation at least as monitored using various lacZ and *GCN4*_lacZ fusion constructs [76,77].

Fortuitously, we discovered that the mRNA encoding the Ure2p protein in yeast possesses an unusual IRES (internal ribosome entry site) located in the open reading frame of the gene [80]. Internal initiation mediated by the *URE2* IRES element was found to lead to the synthesis of an N-terminally truncated form of the protein (amino acids 94-354) [80]. Surprisingly, we found that eIF2A functions as a suppressor of the *URE2* IRES in yeast cells [77] and the regulation of expression from the *URE2* IRES appeared to be dependent on the levels of eIF2A [77,81,82,83]. Moreover, we further found that eIF2A functions as a suppressor of several other yeast IRESs, at least as tested using IRES-mediated initiation of *GIC1* and *PAB1* mRNAs [83].

The exact understanding of the suppression mechanism is lacking. One possible explanation is that eIF2A works as a Met-tRNA_i_ binding protein and directs the binding of Met-tRNA_i_ to 40S ribosomal subunits; however, the rate at which the 48S PIC is converted to an 80S elongation competent complex in presence of eIF2A and the subsequent release of eIF2A from the 80S ribosome is much slower than that for eIF2•GTP•Met-tRNA_i_. The competition between the two mechanisms leads to an apparent suppression of initiation of certain mRNAs in the presence of eIF2A. This mechanism, however, does not explain how exactly eIF2A discriminates between the different mRNAs. Nevertheless, it has been postulated that eIF2A may function with a small subset of mRNAs that rely on alternative initiation mechanisms prevailing under stress conditions when eIF2 becomes inactivated [77]. Interestingly, using a galactose-inducible eIF2A carrying an HA-tag, we were able to show that at least in yeast most of the eIF2A was associated with either 40S or 80S ribosomes [76,77]. This was the first reporting of an initiation factor being associated with the 80S ribosomes, as most other studies found the initiation factors associated with 40S ribosome and polysomes, but not 80S ribosomes. Although yet to be proven, this data could be interpreted as eIF2A being slowly released from initiation complexes (and thereby blocking the subsequent elongation step) and in this manner, while serving to direct Met-tRNA_i_ binding to 40S subunits, would repress expression overall.

## 6. eIF2A Function in Mammalian Cellular Systems

In the mid-2000s, the search for potential eIF2A targets spread to the use of mammalian cell lines and in vitro translation systems [84,85]. Researchers attempted to investigate whether eIF2A might be implicated in the initiation of translation of certain viral and/or cellular mRNAs. In 2006, Iván Ventoso, Luis Carrasco and their co-authors suggested that eIF2A might be involved in the initiation of translation of subgenomic (26S) Sindbis virus (SV) mRNA [84]. SV infection induces PKR activation, which results in a strong phosphorylation (95%) of eIF2α leading to its inactivation [86]. To test the involvement of eIF2A in translation of SV 26S mRNA, the authors silenced the expression of eIF2A by means of siRNA interference and found that abrogation of eIF2A expression led to a reduction in the synthesis of SV structural proteins in PKR^+/+^ cells (that allowed for eIF2α phosphorylation upon viral infection), but not in PKR^0/0^ cells [84]. The effect of eIF2A silencing was restricted to translation of SV 26S mRNA and did not affect translation of genomic mRNAs. The authors concluded that eIF2A is implicated in initiation of SV 26S mRNA [84]. In contrast, a team led by Luis Carrasco subsequently reported a completely opposite observation, suggesting that eIF2A is not required for the translation of SV subgenomic mRNA under conditions of eIF2α phosphorylation [87,88]. In these studies, the authors have used human cell lines where eIF2A was knocked-out by CRISPR/Cas9 genome engineering [87,88].

Further attempts to evaluate cellular eIF2A involvement in minor initiation pathways were made with the use of classical viral IRESs, such as hepatitis C viral (HCV) IRES [89]. In 2011, Sung Key Jang and co-authors investigated the effect of eIF2A depletion on HCV IRES-mediated translation both in cellular and in vitro systems [90]. They found that HCV IRES-dependent translation was not affected by knockdown of eIF2A under normal conditions, but required eIF2A under conditions of eIF2α phosphorylation (both ex vivo in cells and in vitro) [90]. The authors also reported that direct interaction of eIF2A with the IIId domain of the HCV IRES was required for eIF2A-dependent translation [90]. In contrast to these findings, more recent reports from Kieft’s and Carrasco’s labs have shown that depletion of eIF2A in Huh-7 cells had no effect on the translation of HCV mRNA under conditions of extensive eIF2α phosphorylation [91,92]. Thus, the exact function of eIF2A in HCV viral IRES-mediated translation remains to be clarified.

Interestingly, Sung Key Jang has recently reported that eIF2A could be required for the cellular IRES-mediated translation of a non-receptor protein tyrosine kinase c-Src mRNA under conditions of stress [93]. c-Src IRES-dependent translation was found to be greatly reduced by the knockdown of eIF2A under stress conditions caused by tunicamycin treatment [93]. The authors also showed that eIF2A specifically interacts with the c-Src IRES and that mutations in the c-Src IRES that impair this interaction, also abrogate the translation of c-Src mRNA under stress conditions, which lead to eIF2α phosphorylation [93]. Furthermore, using filter-binding assays of [^32^P]-labeled tRNA_i_ the authors demonstrated that eIF2A and c-Src IRES cooperatively facilitate the recruitment of tRNA_i_ onto the 40S ribosomal subunit [93]. Despite the controversial nature of these reports, these studies further suggested that eIF2A may aid the initiation of certain specific mRNAs, and, perhaps, does so in a cell-specific manner.

## 7. eIF2A Function in Mammalian Systems in Non-AUG Initiation Events

Although it was originally thought that eukaryotic translation initiation almost exclusively relied on an AUG start codon, recent advancements in ribosome profiling have revealed that near cognate/non-AUG start codons (like e.g., CUG, GUG, UUG) are also used at a relatively high frequency [54]. These non-AUG initiation events were suggested to be important for regulation of protein synthesis during development and/or stress [54].

In 2012, Nilabh Shastri and co-authors reported that effective immune surveillance by cytotoxic T cells (that requires newly synthesized polypeptides for presentation by major histocompatibility complex (MHC) class I molecules) depends on cryptic non-AUG-initiated translational events that relied on the expression of eIF2A [94]. Selective eIF2A knockdown was shown to significantly inhibit CUG-initiated, but not AUG-initiated, presentation events [94]. In contrast, neither CUG-initiated nor AUG-initiated antigen presentation events were altered by eIF2D knockdown [94]. It was therefore suggested that eIF2A may also enable non-AUG translation in the case of antigen presentation by MHC class I molecules and that eIF2A affects the expression of a specific subset of mRNAs, which are not targets of eIF2D [94]. Yet the exact affinity of eIF2A to, e.g., Leu-tRNA^CUG^ was not determined [94].

Several subsequent reports implicated eIF2A in several non-AUG-initiation events and/or initiation events on unconventional upstream start codons (involving both uAUG and upstream near-cognate start codons, like CUG). In 2014, Yuxin Yin and co-authors found that eIF2A controls the expression of one of the isoforms of the phosphatase and tensin homologue deleted on chromosome 10 (PTEN) protein (which is lost or mutated in many cancers) [95]. PTEN is a powerful tumor suppressor gene, which encodes a 403 amino acid (aa) protein [96]. PTEN has a wide range of biological functions beyond tumor suppression [96]. Yin and co-authors identified the so-called PTENα isoform important for mitochondrial energy metabolism [95]. eIF2A-mediated translation of PTENα was found to be initiated from an upstream CUG codon located in-frame with the start codon of the of canonical PTEN open reading frame (ORF) [95]. This CUG start codon generated an N-terminally extended form of PTEN with an additional 173 aa residues in *H. sapiens* or 169 aa residues in *M. musculus* [96]. Overexpression of eIF2A significantly increased PTENα expression, whereas eIF2A silencing led to PTENα downregulation [95].

In 2016, Nilabh Shastri, Peter Walter, and their co-authors implicated eIF2A as an important player in the integrated stress response (ISR) [97]. ISR triggers eIF2α phosphorylation, which leads to downregulation of cap-dependent protein synthesis [98]. However, protein synthesis does not cease completely on all mRNAs and several mechanisms are known to lead to the recovery from this type of stress [98,99]. The endoplasmic reticulum associated chaperone, binding immunoglobulin protein (BiP), is known to play an important role during stress and was found to be continuously expressed during the ISR [97]. The authors showed that the BiP 5′ UTR harbors uORFs that are exclusively initiated by UUG and CUG start codons and that eIF2A plays a key role in the UUG-initiated uORF translation that appeared to be necessary for BiP expression during the ISR [97]. They concluded that during the ISR cells use an eIF2A-mediated initiation pathway to sustain expression of a cohort of particular proteins [97].

More recently, Jonathan Weissman, Elaine Fuchs and their colleagues showed that eIF2A may be involved in tumor initiation and progression [100]. Translational control is believed to play an important role in cellular transformation and malignancy [101]. Fuchs and colleagues used ribosome profiling to investigate the translational landscape during the transition from normal cellular homeostasis to malignancy [100]. Interestingly, they found that during tumor initiation, the translational apparatus is redirected towards unconventional upstream initiation sites, in particular, involving CUG codons, and that eIF2A is essential for enhancing the translational efficiency from these sites [100]. Silencing of eIF2A led to a substantial decrease in tumor formation in mice engrafted with *Eif2a*-null squamous cell carcinomas (SCCs), and reintroduction of eIF2A into these cell lines, rescued tumor formation [100]. Interestingly, examination of The Cancer Genome Atlas (TCGA) performed by the authors revealed that the *EIF2A* locus is amplified in 29% of human patients with lung SCC, 15% of patients with head and neck SCC and 15% of patients with esophageal carcinoma [100]. The authors thus concluded that eIF2A-mediated translation is important for tumor initiation and progression [100].

Several additional recent reports have shown that eIF2A has specific functions during the ISR as well as cancer progression [102,103]. Chen et al. demonstrated that eIF2A promotes cell survival during paclitaxel-mediated ISR in vitro and in vivo [102]. Paclitaxel is one of the major chemotherapy drugs for breast cancer treatment and its application is associated with strong ISR [102]. The ISR, however, is believed to be critical for cancer cell survival during stress stimuli and has been implicated in the resistance to cancer therapeutics [102]. The authors showed that the loss of ISR increases paclitaxel-mediated cell death and that eIF2A is one of the key and essential factors for cancer cell survival under the conditions of paclitaxel-mediated ISR [102]. Knockdown of eIF2A substantially impaired cancer cell survival under the ISR [102]. The authors also found that elevated levels of eIF2A mRNA in patients with breast cancer are correlated with poor prognosis [102].

Sonobe et al. additionally showed that eIF2A aids translation of the *C9ORF72* mRNA during the ISR stress in neurons and, specifically, helps translation of the expansion of a hexanucleotide repeat (HRE), GGGGCC motifs (encoding poly(Gly Ala) repeats in the *C9ORF72* gene) [103]. Expansion of HREs in the *C9ORF72* gene is recognized as the most common cause of familial amyotrophic lateral sclerosis (FALS) and frontotemporal dementia (FTD) [103]. It was suggested that translation products of *C9ORF72* involving HREs are toxic, may induce ISR and play a critical role in disease pathogenesis [103]. The upstream start codon preceding the repeats that directs the synthesis of poly(Gly Ala) was identified as CUG [103]. Knockout of eIF2A dramatically impaired the synthesis from the *C9ORF71* expanded repeat (GGGGCC) during ISR in neuronal cells [103], therefore suggesting a critical role of eIF2A in repeat associated non-AUG translation disorders [103].

## 8. eIF2A Knockout Mouse

The experiments described above suggested a rather unique role of eIF2A in cellular physiology. We note, however, that in all of these experiments, the researchers used cellular models and studied the events that were affected by eIF2A silencing in a variety of cellular systems. Thus, there was a gap in our understanding of how eIF2A functions in mammalian systems in vivo (on the organismal level) and ex vivo/in vitro (in cells). To fill in this gap and to continue the physical and functional characterization of mammalian eIF2A, we have created a homozygous eIF2A-total knockout (KO) mouse strain [104]. A gene trap cassette was inserted between a very short (28 nt) eIF2A exon 1 and eIF2A exon 2 disrupting expression of all exons downstream of the insertion [104]. The eIF2A knockout mice appeared to be viable and fertile and exhibited no breeding abnormalities under standard growth conditions further suggesting that eIF2A is not involved in key initiation pathways and vital for organismal functions [104]. Interestingly, our analysis of eIF2A protein expression in six different organs (heart, brain, lung, liver, kidney, pancreas) in wild-type mice showed that the highest eIF2A protein abundance was observed in mouse pancreas and liver and that the eIF2A protein expression varied substantially between different tissues [104]. We could not detect any protein in kidneys (of the wild-type animal) even though a similar amount of eIF2A mRNA was present in this tissue in comparison with e.g., liver and pancreas [104]. This suggests that the eIF2A mRNA may be translationally silenced in kidneys and that eIF2A may be itself a subject of translational control [104]. Thus, it seems that eIF2A is not a ubiquitously expressed factor (at least on the protein level) in contrast to eIF2 [61,62]. Several estimates have been made suggesting that the abundance of eIF2A in yeast and HeLa cells is comparable to that of eIF5B and about 3-fold lower than that for eIF2 [57].

Initially we found that the eIF2A KO mice revealed no visible phenotype (at about 3–5 months of age), being similar in size and morphology to wild-type mice [104]. However, further in-depth analysis of these mice has shown that eIF2A affects a variety of physiological and pathophysiological processes in mice and in particular its absence leads to the development of a metabolic syndrome (in preparation).

## 9. eIF2A Structure

As noted above, eIF2A protein homologues are found in a wide range of eukaryotic species, suggesting a conserved biological role [76,77]. However, the evolutionary origin of the eIF2A is not quite clear yet, as no homologues have been found in prokaryotes and archea. eIF2A was predicted to harbor at least 2 domains N- and C-terminal [76,77]. eIF2A N-terminal part was predicted to adopt a WD-repeat β-propeller fold, the structure of which (for yeast *Schizosaccharomyces pombe* eIF2A protein) has been solved at 2.5Å resolution [105]. Interestingly, the C-terminally truncated *S. pombe* eIF2A fragment (residues 1-424) adopts an unconventional 9-bladed β-propeller fold [105] with the same overall topology as many other β-propeller proteins [106,107]. Homology modeling of the human eIF2A (Figure 4) suggests that human eIF2A N-terminal domain (residues 1-415) may also adopt the same 9-bladed β-propeller fold and that the C-terminal domain (residues 416-585) may be less structured and, perhaps, on its own consist of 2-3 smaller subdomains (with very C-terminal fragment residues 533-585 represented by a helix-bundle) (Figure 4).

WD repeat β-propeller domains play an important role in protein–protein interactions [106,107]. Several ribosome-binding proteins (e.g., RACK1 [108] and eIF3b [109] (a subunit of eIF3)) are known to contain β-propeller domains. The eIF3b structure of the filamentous fungus, *Chaetomium thermophilum,* has been solved and was also found to contain a 9-bladed β-propeller fold [109] that was suggested to drive the association of eIF3b with the ribosome [109]. Apart from 40S subunit binding; eIF3b was suggested to play important roles in multisubunit eIF3 assembly; TC and mRNA recruitment as well as scanning [9]. Interestingly, WD repeat domains of eIF2A and eIF3b share 21% identity and 40% similarity [76]; however, it is not clear whether eIF2A might have evolved from eIF3b, or vice versa.

The WD repeat β-propeller domain of eIF2A was recently suggested to be responsible for eIF2A Met-tRNA_i_ binding properties, whereas the central part of the eIF2A C-terminal domain (residues 462–501) was suggested to drive the protein’s interaction with eIF5B [79]. This very C-terminal helical subdomain was suggested to be responsible for eIF2A interaction with certain specific mRNAs [79]. It was generally proposed that eIF2A interacts and functionally cooperates with eIF5B during Met-tRNA_i_ loading onto 40S ribosomal subunits specifically under stress [79].

At the same time, a study by Thakor and colleagues [110] has showed that under stress eIF5B cooperates with eIF1A and eIF5, but not eIF2A. This study suggested that eIF5B may play a role in the uORF-mediated regulation of activating transcription factor 4 (ATF4) translation [110]. The ATF4 mRNA is well known to be more efficiently translated under a variety of stress conditions that lead to eIF2α-phosphorylation [111]. The molecular mechanism driving ATF4 translation in response to eIF2α-phosphorylation is similar to that for yeast *GCN4* [112] and relies on translation re-initiation involving uORFs [111].

## 10. eIF2A Interacting Partners

Genetic, biochemical, tandem affinity purification, co-immunoprecipitation and direct binding assays implicated eIF2A in a diverse network of interactions [76,77,78,79,83,113,114,115]. Interacting proteins included, for example, ribosomal proteins from the small and large ribosomal subunits, initiation factors eIF4E and eIF5B, elongation factor eEF1A, several ribosome-associated chaperones and many other proteins [76,77,78,79,83,113,114,115]. Some of these interactions were RNA-dependent, while the others were not [83]. As has been mentioned above, both genetic and physical interactions were observed for initiation factors eIF4E and eIF5B [76,77,78,79], therefore providing additional evidence for eIF2A participation in the initiation process. As noted above, the direct interaction between eIF2A and eIF4E was demonstrated using pull down assays of either tagged eIF2A or tagged eIF4E [79]. The driving force for this study was the observation of the sequence, _446_AYxPPxxR, in the human eIF2A protein, which was found to be similar to a motif previously suggested to drive interaction of DDX3 helicase with eIF4E [78]. This motif is well conserved and is seen in eIF2A sequences from humans to chickens, fish, fruit flies, sea squirt and aspergillus [78].

The amino acids 460–571 in the C-terminal domain of yeast eIF2A were shown (using similar pull down assays) to be responsible for the interaction between eIF2A and eEF1A [83]. Deletion of this region completely abolished the interaction between the two proteins, abrogating eIF2A-mediated repression of internal initiation of the *URE2* IRES [83]. It was, however, unclear to what extent the eIF2A-eEF1A interaction per se is critical for the eIF2A-mediated repression of the *URE2* IRES element, since the deletion of this region could have also affected eIF2A function independent of its interaction with eEF1A [83].

At present, it is not clear to what extent the eIF2A binding partners influence its activity and stability/turnover and this remains to be established. Interestingly, eEF1A has been characterized as an E3 ligase essential for ubiquitin-dependent degradation of certain N alpha-acetylated proteins [116]. We and others have found that eIF2A is N-terminally acetylated [22,76]. Acetylation is the most common co-translational protein modification [117,118]. However, despite its prevalence, the exact biological role of N-terminal acetylation is unknown [117,118], and it is likely to have pleiotropic effects. Whether N-terminal acetylation of eIF2A is important for its biological function remains unknown.

## 11. eIF2A Modifications

eIF2A is often confused with the eIF2α subunit of the eukaryotic initiation factor 2. The primary reason for this confusion is the failure to distinguish between the Greek “α”, the lower case “a” and capital “A”. At present, the UniProtKB database entry for the human eukaryotic translation initiation factor 2 subunit 1 (α) (eIF2α), P05198 (IF2A_HUMAN) mentions eIF-2A as one of the alternative/short names for eIF2α. As such, it often inappropriately stated that eIF2A can undergo phosphorylation on Ser51 in response to stress. We have to unambiguously say that this statement is incorrect.

We note, however, that large scale quantitative phosphoproteomic analyses [119,120,121] do reveal that human eIF2A can undergo phosphorylation on several of its Ser and Thr residues. With the exception of Thr5, most of these modified residues (Ser503, Ser506, Ser517, Thr518, Ser526) are localized in the C-terminus of the protein [119,120,121]. Interestingly, these C-terminal phosphorylated residues are located close to, or within the predicted PEST motif (eIF2A: _505_KSPDLAPTPAPQSTPR). PEST sequences in proteins are regions rich in proline (P), glutamic acid (E), aspartic acid (D), serine (S), and threonine (T) usually confined by two positively charged amino acids, lysine (K), arginine (R) or histidine (H) and were suggested to be responsible for the controlled protein turnover and degradation [122,123]. PEST motifs appeared to function as anchor sites of E3 ubiquitin ligases [124,125] and their binding activity was suggested to be regulated via phosphorylation [123]. We have previously suggested that the PEST motif in the C-terminus of eIF2A may control its turnover [77]. However, whether this is indeed the case and whether phosphorylation of this region can affect eIF2A activity and/or degradation remains to be established. It has to be noted that additional/alternative functions have also been attributed to PEST motifs, like intracellular sorting and/or binding of the SUMO conjugating proteins [126,127,128].

## 12. Alternate Initiation Pathways as a Response to Stress and Non-eIF2-Mediated Translation

To date, the predominant pathway for protein synthesis initiation in eukaryotes is as illustrated in Figure 1. For a long time, this “cap-dependent” or “scanning” mode of initiation was considered the only possible route through which translation of eukaryotic mRNAs could be initiated. However, studies of viral gene expression in the late 1980s led to the discovery of an alternative mode of translation initiation in eukaryotic cells [129,130], a mode that became known as the IRES-mediated pathway [3,48,50,51,55,58] and that bypasses the requirement for cap-dependent scanning. As such, IRES-driven translation appeared to have a generally reduced requirement for canonical translation initiation factors, particularly members of the eIF4F complex (initiation factors eIF4E and eIF4G) required for recognition of the 5′-end cap [3,48,50,51,55,58]. The use of this alternate initiation pathway has been as a result of stress associated with viral infection that is diminishing the “normal” pathway by several mechanisms that commonly lead to inactivation of eIF4E and/or eIF4G or both, as well as the reduction in activity of TC (as a result of targeted phosphorylation of eIF2) [3,48,50,51,55,58].

Further studies of the IRES-mediated pathway have shown that the involvement of canonical initiation factors in IRES-driven translation initiation varies for IRESs in different mRNAs and that in certain “extreme” cases, initiation can proceed without involvement of any of the canonical initiation factors [131]. Additionally, the study of many physiological and pathophysiological stress conditions that lead to inhibition of cap-dependent translation (such conditions apart from viral infection, include, but are not limited to, endoplasmic reticulum (ER) stress, hypoxia, nutrient limitation, mitosis, cellular transformation and differentiation, etc.) have led to the realization that a variety of minor initiation pathways can operate (in case of both viral and cellular mRNAs) in cells under these conditions [3,48,50,51,55,58]. The visualization of the alternate mechanism reflects the “weakness” of their ability to drive initiation when forced to compete with the normal cap-dependent pathway and although these alternate pathways are described as “not competitive”, in aggregate, there are some circumstances where prolonged stress still allows recovery of protein synthesis to more than 50% of the non-stressed condition [99].

A common outcome of many physiological and pathophysiological stress conditions is the phosphorylation of eIF2 [63,64]. The observation of pathways operating under eIF2α phosphorylation has led to an understanding that “eIF2-less initiation” may be more prevalent than was originally thought [51]. In addition, while eIF2A has been the most noted factor believed to be involved in non-eIF2 initiation events (Table 4), other proteins, as discussed above, Ligatin/eIF2D [59,60], MCT-1/DENR) [59] and eIF5B also seem to be able to serve a similar role under limited conditions.

One of the more curious examples is eIF5B, which by sequence, is homologous to bacterial IF2 (the factor that binds the initiator fMet-tRNA_i_) [49,74,75]. An early report indicated that eIF5B was responsible for initiation using the classical swine fever virus (CSFV) IRES element, a reaction that was dependent on domain II of the IRES [49]. HCV IRES was also shown to use the same unconventional pathway in which the 48S complex is formed in an eIF2-independent manner by cooperative Met-tRNA_i_ and eIF5B binding [75]. Thus, it has been suggested that HCV IRES can operate with both eIF2A [90] and eIF5B [75] in place of eIF2. Recently, in a study of oncolytic Maraba virus infection, it was observed that eIF5B was required for the efficient expression of Bcl-xL, a protein required for the optimal propagation of the virus [132]. eIF5B also appears to be involved in several other circumstances of cellular messenger RNA translation at least as seen in the case of the X-chromosome linked Inhibitor of Apoptosis, XIAP, mRNA [133].

As has also been mentioned previously, one of the other/alternate factors that can bind Met-tRNA_i_ is eIF2D [59,60], and a complex of MCT-1/DENR [59]. In the original description of the protein, eIF2D had a quite different degree of specificity for tRNAs being able to bind both Met-tRNA_i_, Met-tRNA_m_ or other non-acylated tRNAs to the 40S ribosomal subunits [59,60]. This was in contrast to eIF2A and eIF5B, which appeared to be more specific for initiator tRNA_i_-like looking species (Met-tRNA_i_ and Phe-tRNA*^E.coli^*) (Table 2). However, as noted above, it is possible that eIF2A also recognizes Leu-tRNA.

Recent reports implicated eIF2D and MCT-1/DENR in re-initiation events downstream of uORFs [52,134,135,136]. The control of these events by MCT-1/DENR in particular appeared to be important for the control of cell proliferation and protein synthesis in proliferating but not quiescent cells in *Drosophila* [134]. Moreover, it appeared that DENR depletion can shorten circadian period in mouse fibroblasts, suggesting involvement of uORF usage and re-initiation in circadian/clock regulation [135].

Translatome and transcriptome analysis of Tma64/eIF2D, Tma20/MCT-1, and Tma22/DENR knockout yeast strains further provided the basis for genome wide analysis of the role of these factors in the stress response, mating and sporulation in *S. cerevisiae* and the identification of specific mRNAs whose re-initiation or de novo initiation has been influenced by gene deletions [136,137]. Finally, structural studies of the eIF2D and the complex of MCT-1/DENR bound to the 40S ribosomal subunit in complex with initiator tRNA (using a combination of cryoelectron microscopy and X-ray crystallography, or X-ray crystallography alone) revealed how these proteins function and bring the initiator tRNA to the ribosomal P-site [138,139]. Similar studies and understanding are lacking in the case of eIF2A.

## 13. Conclusions

The discovery that several factors can function in place of eIF2 presents a new paradigm for the understanding of the mechanism of translation initiation and its regulation. A largely unexplored area of the research, which is devoted to analysis of the eIF2A function in cellular physiology, integrated stress response and human disease, has recently attracted substantial attention. It is becoming increasingly clear that eIF2A preferentially participates in translation of a specific subset of mRNAs, most likely involving upstream non-canonical initiation codons (uCUG, uUUG) and is supporting the translation of these mRNAs under conditions of ISR (eIF2α phosphorylation) (Figure 5).

It is not yet clear, however, how the selection between different “minor” proteins that are able to bind Met-tRNA_i_ (eIF2A, eIF2D, eIF5B, MCT-1/DENR) is achieved and how their targets are identified by the cellular protein synthesis machinery. It is also not clear how exactly the assembly of the 48S complex is achieved in terms of the order of binding of eIF2A, mRNA and the 40S ribosomal subunit and different scenarios have been proposed [51]. Therefore, understanding the effects of eIF2A on selective protein synthesis and its role in disease development (such as, e.g., cancer) is extremely important, as it may further provide an experimental platform to aid in the design and discovery of novel therapeutics.

Undoubtedly, a combination of in vivo and ex vivo approaches will allow many important questions to be addressed related to eIF2A function. However, determination of the exact mechanism of eIF2A action will likely require the use of fully reconstituted initiation systems and in-depth structural studies of eIF2A bound ribosomal complexes.

## Figures and Tables

**Figure 1 ijms-21-02054-f001:**
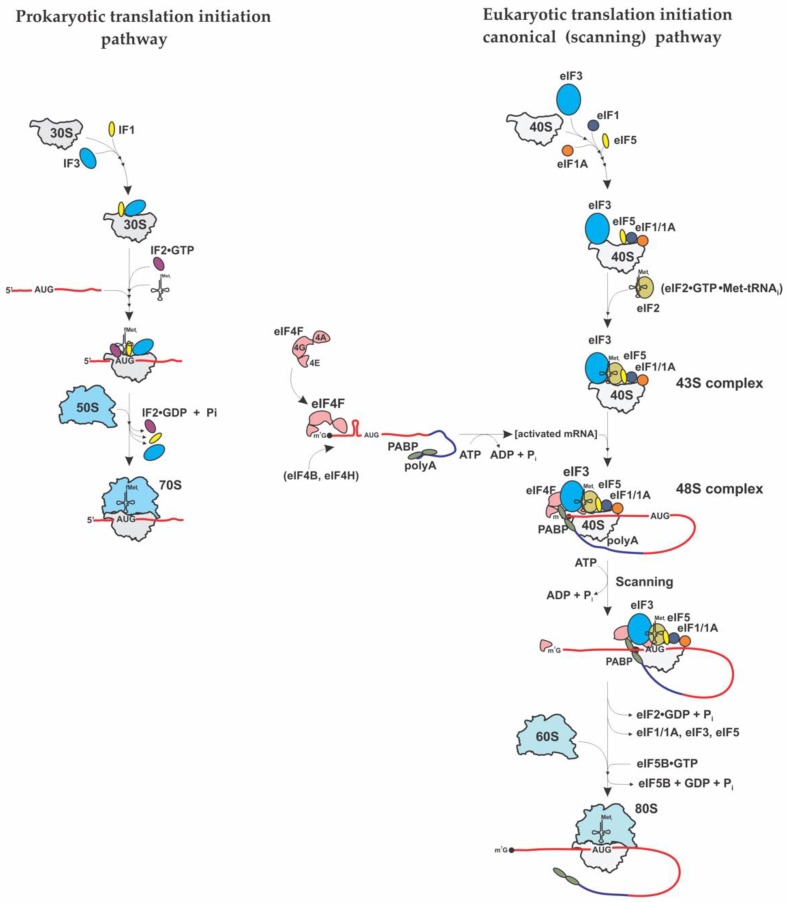
Overview of the major stages of prokaryotic (left) and eukaryotic cap-dependent translation (right) initiation pathways.

**Figure 2 ijms-21-02054-f002:**
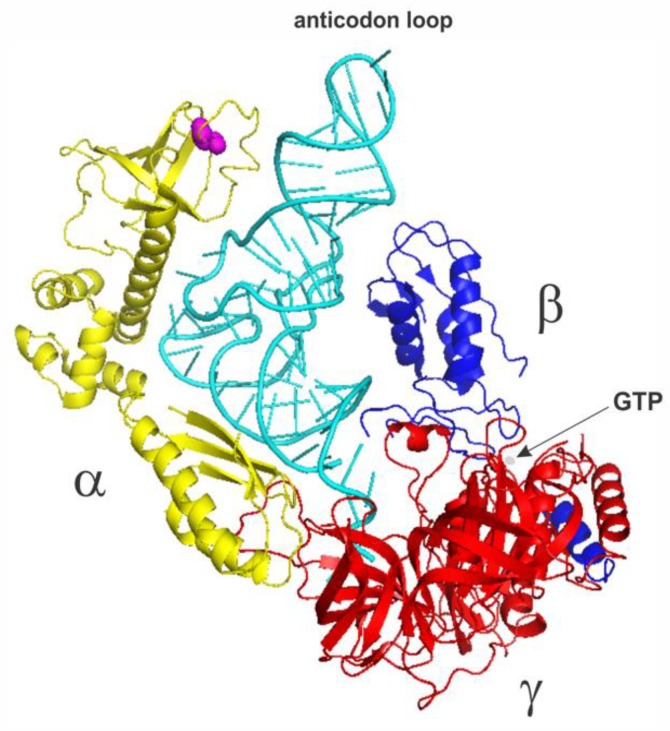
The structural model of the yeast eIF2•GTP•Met-tRNA_i_ ternary complex adapted from the structure of the yeast 48S complex (PDB 3JAP [65]) with the 40S ribosome and other factors omitted. The eIF2α subunit is in yellow (Ser51 is in magenta; Van der Waals radii of the side chain atoms are shown), the eIF2β subunit is in blue and the eIF2γ subunit is in red. Met-tRNA_i_ is in cyan. Arrow points to GTP (in gray).

**Figure 3 ijms-21-02054-f003:**
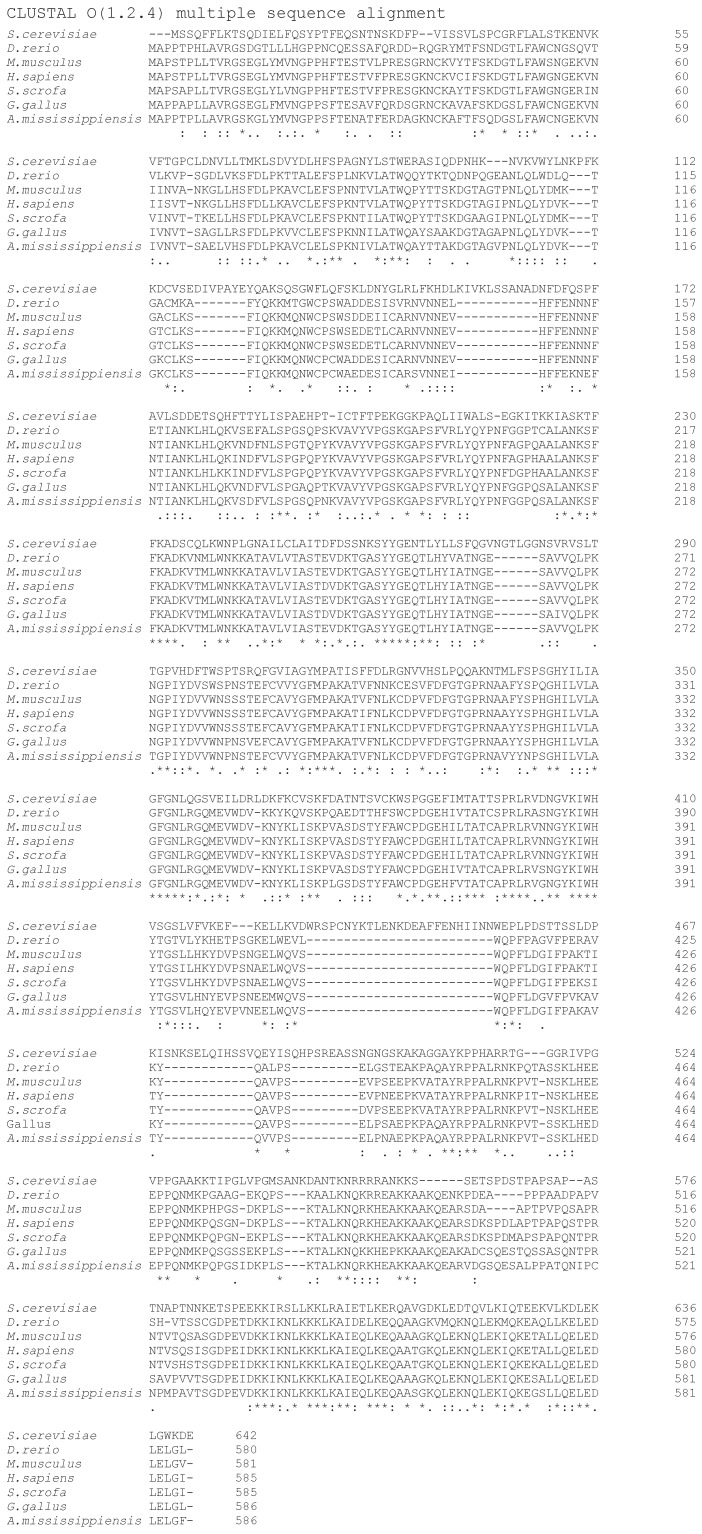
Representative sequence alignments of eIF2A proteins from yeast (*Saccharomyces cerevisiae*), zebrafish, (*Danio rerio*), mouse (*Mus musculus*), human (*Homo sapiens*), pig (*Sus scrofa*), chicken (*Gallus gallus*) and alligator (*Alligator mississippiensis*).

**Figure 4 ijms-21-02054-f004:**
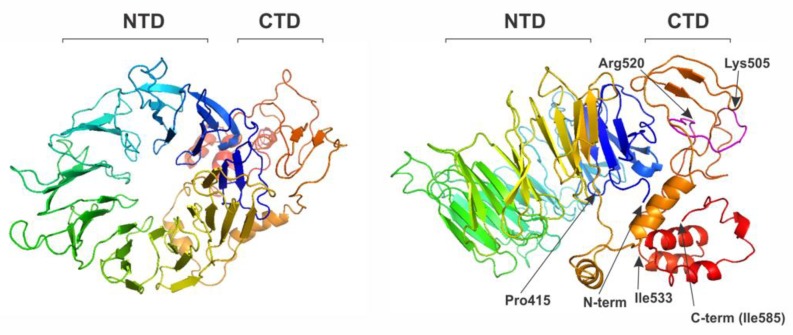
Predicted structure of the human eIF2A protein, showing top view (left) and side view (right) of the β-propeller domain. N-terminal domain (NTD) (residues 1-415) folds into 9-bladed β-propeller. The C-terminal domain (CTD) is less structured and may consist of two to three smaller subdomains (with very C-terminal fragment residues 533-585 represented by a helix bundle). Residues 505–520 (in magenta on the right image) represent a potential PEST motif. Homology modeling was done using Phyre2 Protein Fold Recognition Server.

**Figure 5 ijms-21-02054-f005:**
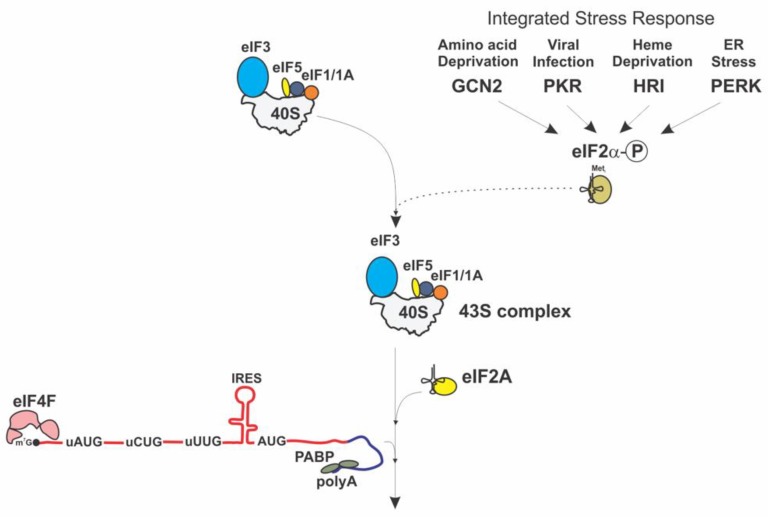
eIF2A initiation pathway (only the stage leading to the assembly of 48S complex is shown).

**Table 1 ijms-21-02054-t001:** Eukaryotic factors initially identified to be capable of binding Met-tRNA_i_ to the 40S subunit.

Merrick/Anderson Rabbit Reticulocytes Ref. [22]	Ochoa *A. salina* Ref. [19]	Hardesty Rabbit Reticulocytes Ref. [25]	Moldave Rat Liver Ref. [18]
IF-M1 MW ~65,000 Da	aIF-2 MW ~(74,000)_2_ Da	Binding factor MW ~50,000 Da (~30,000+20,000 Da)	RL IF-1 MW ~70,000 Da

**Table 2 ijms-21-02054-t002:** tRNA binding properties of eIF2A (upon identification), eIF2D, MCT-1/DENR and eIF5B.

	Factor	eIF2A	eIF2D	MCT-1/DENR	eIF5B
Assay		Ref. [22,24]	Ref. [59,60]	Ref. [59,60]	Ref. [49,74,75]
**codon-directed** **binding to 40S**		Met-tRNA_i_ Phe-tRNA*^E.coli^*	Met-tRNA_i_ Phe-tRNA tRNA^Met^_i_ (non-acylated)	Met-tRNA_i_ Phe-tRNA	Met-tRNA_i_
**aa-puromycin**		Met-tRNA_i_			
**poly(U)-directed synthesis** **of polyphenylalanine**		Phe-tRNA*^E.coli^*		Phe-tRNA	Phe-tRNA*^E.coli^*
**toe-printing**			Met-tRNA_i_	Met-tRNA_i_	Met-tRNA_i_

**Table 3 ijms-21-02054-t003:** Amino acid composition of rabbit IF-M1/eIF2A and eIF2D.

IF-M1 Rabbit Ref. [22] Number of aa: 585			eIF2A Rabbit (UNIPROT: G1TAW7) Number of aa: 585			eIF2D Rabbit (UNIPROT: P0CL18) Number of aa: 566		
MW: ~65000.00 pI: ND in Ref. [22]			MW (predicted): 64839.70 Theoretical pI: 9.03			MW (predicted): 62208.50 Theoretical pI: 6.97		
Ala (A)	42.6	7.3%	Ala (A)	45	7.7%	Ala (A)	40	7.1%
Arg (R)	20.3	3.5%	Arg (R)	13	2.2%	Arg (R)	22	3.9%
Asn (N)			Asn (N)	35	6.0%	Asn (N)	15	2.7%
Asp (D)	54.3	9.3%	Asp (D)	23	3.9%	Asp (D)	30	5.3%
Cys (C)	9	1.5%	Cys (C)	9	1.5%	Cys (C)	11	1.9%
Gln (Q)			Gln (Q)	27	4.6%	Gln (Q)	36	6.4%
Glu (E)	61.4	10.5%	Glu (E)	32	5.5%	Glu (E)	34	6.0%
Gly (G)	38.9	6.6%	Gly (G)	36	6.2%	Gly (G)	37	6.5%
His (H)	12.2	2.1%	His (H)	12	2.1%	His (H)	16	2.8%
Ile (I)	22.4	4 3.8%	Ile (I)	22	3.8%	Ile (I)	20	3.5%
**Leu (L)**	**46**	**7.9%^1^**	**Leu (L)**	**46**	**7.9%**	**Leu (L)**	**68**	**12.0%**
**Lys (K)**	**61.4**	**10.5%**	**Lys (K)**	**55**	**9.4%**	**Lys (K)**	**41**	**7.2%**
Met (M)	7.9	1.3%	Met (M)	7	1.2%	Met (M)	10	1.8%
**Phe (F)**	**19.4**	**3.3%**	**Phe (F)**	**24**	**4.1%**	**Phe (F)**	**12**	**2.1%**
Pro (P)	40.4	6.9%	Pro (P)	45	7.7%	Pro (P)	39	6.9%
Ser (S)	42.7	7.3%	Ser (S)	45	7.7%	Ser (S)	40	7.1%
Thr (T)	36.9	6.3%	Thr (T)	39	6.7%	Thr (T)	29	5.1%
**Trp (W)**	**9.3**	**1.6%**	**Trp (W)**	**11**	**1.9%**	**Trp (W)**	**5**	**0.9%**
Tyr (Y)	16.2	2.7%	Tyr (Y)	22	3.8%	Tyr (Y)	13	2.3%
Val (V)	44.4	7.6%	Val (V)	37	6.3%	Val (V)	48	8.5%

^1^ Characteristic amino acid residues Leu, Lys, Phe and Trp, showing substantially dissimilar usage in rabbit IF-M1/eIF2A and eIF2D are highlighted in bold type.

**Table 4 ijms-21-02054-t004:** eIF2A participation in initiation pathways.

ProcessEffect	IRES-MediatedTranslation	Re-Initiation	Standard AUGand uAUGTranslation	uCUG, uUUGTranslation
Positive—(eIF2A facilitates)	HCV: Ref. [90]s-Src kinase: Ref. [93]	BiP (potentially):Ref. [97]	Sindbis virus (SV): Ref. [84].Oncogenes: Ref. [100]	MHC I-peptides: Ref. [94]PTENα: Ref. [95]BiP: Ref. [97]Oncogenes: Ref. [100,102]*C9ORF72*: Ref. [103]
Negative—(eIF2A suppresses)	*URE2, GIC1, PAB1*: Ref. [77,81,82,83]			
Does not affect	HCV: Ref. [91,92]	*GCN4*: Ref. [76]	SV: Ref. [87,88]

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
