# Peer review of "A Retrospective on eIF2A—and Not the Alpha Subunit of eIF2"

_ijms, 2020, doi:10.3390/ijms21062054_

Round 1
Reviewer 1 Report
eIF2A is one of first components of the eukaryotic translation initiation process to be purified. Research on eIF2A languished when it was found not to function in the major steps in the initiation process. This manuscript represents a timely review of eIF2A as being important for translational control of specific mRNAs. The article is helpful in placing eIF2A in its historic context. The language is sometimes a bit impenetrable and suggestions have been made to simplify the text to increase access by non-oficionados. Questions arising: what is the origin of eIF2A? What is the expression level of eIF2A compared to the other initiation factors?
Abstract, line 23: change to “but is suggested”
Line 47: The 30S subunit then binds to …
Line 65: post-termination. As well It also plays ..
Line 67: eIF3 also promotes 43 S complex binding to the 5’-end of mRNA via a bridge with eIF4G (a part of the eIF4F complex (see below) and 40S-bound eIF3) to initiate further scanningof the 43S complex.
Line 83: to form the multisubunit eIF4F complex
Line 89: largely triggered guided by
Line 93: The ability to inherit these apply such assays was..
Lines 102, 111 & 112: mid-19 70’s and 1980’s
Line 115: research on eIF2A seemed to have been abandoned languished.
Line 120: oncogene product, MCT-1, and
Line 123: eIF2 is rendered inactive less active.
Lines 122 an 123 – this sentence is a non-sequitor, something is missing.
Line 126: suggest “ While eIF2A was the factor first described as capable…”
Line 128: ..this factor is a the key player..
Line 128: and binds the guanine nucleotides, GTP and GDP and ..
Line: 149: ..is affected by the nature of guanine nucleotides.
Line 150: Following the location placement of 40S..
Line 168: by eIF4E the eIF4F complex
Line 181: was triggered guided by
Line 183: sentence is impenetrable. Suggest instead “Unlike in bacteria, mRNA interaction with the 40S subunit in eukaryotes only occurs after Met-tRNAi binding to the partial P-site. Because of this, it was not understood that eIF2 is capable of binding Met-tRNAi to the 40S ribosomal subunit in a GTP-dependent, but AUG-independent manner (refs).
Line 188: complex would requires GTP,
Line 209: identification of eIF2A (at that time called IF-M1) involved preparation
Line 225: As such From these studies, a non-IF-M1
Line 233: These experiments also yielded several preparations of eIF2A
Line 234: eIF2A preparations isolated in this later study [60] did not appear to possess
Line 250: The subsequent identification of eIF2A partial protein and complete gene sequences [76] The identification of the cDNA of human eIF2A and its predicted amino acid sequence has unambiguously shown…. (GenBankTM accession number AF497978).
Line 255: In depth analysis of eIF2A undoubtedly required was facilitated by
Line 302: At that time however we were fortunate to Fortuitously, we discovered
Line 307: appeared to be dependent on through the levels of eIF2A [refs]. Protein
Line 310: the suppression mechanism is as lacking
Line 315: the two mechanisms leads to an apparent suppression of initiation of certain mRNAs in the presence of eIF2A presence for
Line 330: In the mid-2000’s, the search for potential eIF2A targets has been spread to the use of mammalian cell lines and in vitro translation systems cellular systems [84, 85].
Line 341: Curiously In contrast, a team led by Luis Carrasco
Line 343: This time have the authors In these studies, the authors used human cell lines where in whicheIF2A was knocked-out by..
Line 386: As such, In 2014, Yuxin Yin and co-authors have found that eIF2A controls the expression of one of the isoforms of the a phosphatase and tensin homologue deleted on chromosome 10 (PTEN) protein
Line 390: However, PTEN has , however, also a wide range of biological functions
Line 402: The E endoplasmic reticulum associated chaperone, binding immunoglobulin
protein (BiP), is known
Lines 423-427, very confusing sentence
line 437: characterization of a mammalian eIF2A
line 440: The eIF2A knockout mice
line 442: in key initiation pathways and vital for organismal functions
line 456: yeast to humans is not a wide phylogenetic spread
line 482: was suggested to drive the protein’s interaction with eIF5B
line 510: as well as it abrogating ed eIF2A-mediated
line 513: since it was suggested that the deletion of this region could have also affected eIF2A function
lines 618-620: very confusing sentence
lines 626-629: will allow many important questions one to be addressed related to eIF2A function in living systems
Author Response
We are very grateful to the reviewers for their careful reading of our manuscript and for their insightful comments and suggestions.
Referee #1 major points:
Refeere #1: “eIF2A is one of first components of the eukaryotic translation initiation process to be purified. Research on eIF2A languished when it was found not to function in the major steps in the initiation process. This manuscript represents a timely review of eIF2A as being important for translational control of specific mRNAs. The article is helpful in placing eIF2A in its historic context. The language is sometimes a bit impenetrable and suggestions have been made to simplify the text to increase access by non-oficionados. Questions arising: what is the origin of eIF2A? What is the expression level of eIF2A compared to the other initiation factors?”
Response: The reviewer raises very interesting questions that we attempted to address on pages 14, 15 and 16 of the revised manuscript.
Regarding the origin of eIF2A, we state the following:
(lines 471-473): “As noted above eIF2A protein homologues are found in a wide range of eukaryotic species, suggesting a conserved biological role [76,77]. However, the evolutionary origin of the eIF2A is not quite clear yet, as no homologues have been found in prokaryotes and archea.”
(lines 493-496): “Interestingly, WD repeat domains of eIF2A and eIF3b share 21% identity and 40% similarity [76], however it is not clear whether eIF2A might have evolved from eIF3b, or vise versa.”
With regard to the eIF2A expression levels, we state the following:
(lines 453-456): “Interestingly, our analysis of eIF2A protein expression in six different organs (heart, brain, lung, liver, kidney, pancreas) in wild-type mice showed that the highest eIF2A protein abundance was observed in mouse pancreas and liver and that the eIF2A protein expression varied substantially between different tissues [105].”
(lines 460-463): Thus, it seems that eIF2A is not a ubiquitously expressed factor (at least on the protein level) in contrast to eIF2 [61,62]. A number of estimates have been made suggesting that the abundance of eIF2A in yeast and HeLa cells is comparable to that of eIF5B and about 3-fold lower than that for eIF2 [57].
Referee #1 minor issues:
Abstract, line 23: change to “but is suggested”
Response: The text has been changed as suggested by the reviewer.
Line 47: The 30S subunit then binds to …
Response: The text has been changed as suggested by the reviewer.
Line 65: post-termination. As well It also plays ..
Response: The text has been changed as suggested by the reviewer.
Line 67: eIF3 also promotes 43 S complex binding to the 5’-end of mRNA via a bridge with eIF4G (a part of the eIF4F complex (see below) and 40S-bound eIF3) to initiate further scanningof the 43S complex.
Response: The text has been changed as suggested by the reviewer.
Line 83: to form the multisubunit eIF4F complex
Response: The text has been changed as suggested by the reviewer.
Line 89: largely triggered guided by
Response: The text has been changed as suggested by the reviewer.
Line 93: The ability to inherit these apply such assays was..
Response: The text has been changed as suggested by the reviewer.
Lines 102, 111 & 112: mid-19 70’s and 1980’s
Response: The text has been changed as suggested by the reviewer.
Line 115: research on eIF2A seemed to have been abandoned languished.
Response: The text has been changed as suggested by the reviewer.
Line 120: oncogene product, MCT-1, and
Response: The text has been changed as suggested by the reviewer.
Line 123: eIF2 is rendered inactive less active.
Response: The text has been changed as suggested by the reviewer.
Lines 122 an 123 – this sentence is a non-sequitor, somethingismissing.
Response: The text (lines 116-120) has been modified as follows:
“In addition, it appeared that several other factors, and, specifically, Ligatin/eIF2D [59,60] and the complex of the oncogene product, MCT-1, and density regulated protein (DENR) [59] can promote recruitment of Met-tRNAi to some 40S/mRNA complexes under conditions of inhibition of eIF2 activity. Thus, it became clear that factors like eIF2A may affect the translation of only a subset of mRNAs and this may happen under conditions when eIF2 is rendered less active.”
Line 126: suggest “ While eIF2A was the factor first described as capable…”
Response: The text has been changed as suggested by the reviewer.
Line 128: ..this factor is a the key player..
Response: The text has been changed as suggested by the reviewer.
Line 128: and binds the guanine nucleotides, GTP and GDP and ..
Response: The text has been changed as suggested by the reviewer.
Line: 149: ..is affected by the nature of guanine nucleotides.
Response: The text has been changed as suggested by the reviewer.
Line 150: Following the location placement of 40S..
Response: The text has been changed as suggested by the reviewer.
Line 168: by eIF4E the eIF4F complex
Line 181: was triggered guided by
Response: The text has been changed as suggested by the reviewer.
Line 183: sentence is impenetrable. Suggest instead “Unlike in bacteria, mRNA interaction with the 40S subunit in eukaryotes only occurs after Met-tRNAi binding to the partial P-site. Because of this, it was not understood that eIF2 is capable of binding Met- tRNAi to the 40S ribosomal subunit in a GTP-dependent, but AUG-independent manner (refs).
Response: The text has been changed as suggested by the reviewer.
Line 188: complex would requires GTP,
Response: The text has been changed as suggested by the reviewer.
Line 209: identification of eIF2A (at that time called IF-M1) involved preparation
Response: The text has been changed as suggested by the reviewer.
Line 225: As such From these studies, a non-IF-M1
Response: The text has been changed as suggested by the reviewer.
Line 233: These experiments also yielded several preparations of eIF2A
Response: The text has been changed as suggested by the reviewer.
Line 234: eIF2A preparations isolated in this later study [60] did not appear to possess
Response: The text has been changed as suggested by the reviewer.
Line 250: The subsequent identification of eIF2A partial protein and complete gene sequences [76] The identification of the cDNA of human eIF2A and its predicted amino acid sequence has unambiguously shown…. (GenBankTM accession number AF497978).
Response: The text has been changed as suggested by the reviewer.
Line 255: In depth analysis of eIF2A undoubtedly required was facilitated by
Response: The text has been changed as suggested by the reviewer.
Line 302: At that time however we were fortunate to Fortuitously, we discovered
Response: The text has been changed as suggested by the reviewer.
Line 307: appeared to be dependent on through the levels of eIF2A [refs]. Protein
Response: The text has been changed as suggested by the reviewer.
Line 310: the suppression mechanism is as lacking
Response: The text has been changed as suggested by the reviewer.
Line 315: the two mechanisms leads to an apparent suppression of initiation of certain mRNAs in the presence of eIF2A presence for
Response: The text has been changed as suggested by the reviewer.
Line 330: In the mid-2000’s, the search for potential eIF2A targets has been spread to the use of mammalian cell lines and in vitro translation systems cellular systems [84, 85].
Response: The text has been changed as suggested by the reviewer.
Line 341: Curiously In contrast, a team led by Luis Carrasco
Response: The text has been changed as suggested by the reviewer.
Line 343: This time have the authors In these studies, the authors used human cell lines where in which eIF2A was knocked-out by..
Response: The text has been changed as suggested by the reviewer.
Line 386: As such, In 2014, Yuxin Yin and co-authors have found that eIF2A controls the expression of one of the isoforms of the a phosphatase and tensin homologue deleted on chromosome 10 (PTEN) protein
Response: The text has been changed as suggested by the reviewer.
Line 390: However, PTEN has , however, also awiderangeofbiologicalfunctions
Response: The text has been changed as suggested by the reviewer.
Line 402: The E endoplasmic reticulum associated chaperone, binding immunoglobulin protein (BiP), is known
Response: The text has been changed as suggested by the reviewer.
Lines 423-427, very confusing sentence
Response: The text was revised and now reads as follows (Lines 420-440):
A number of additional recent reports showed that eIF2A has specific functions during the ISR as well as cancer progression [103,104]. Chen et al. demonstrated that eIF2A promotes cell survival during paclitaxel-mediated ISR in vitro and in vivo [103]. Paclitaxel is one of the major chemotherapy drugs for breast cancer treatment and its application is associated with strong ISR [103]. The ISR, however, is believed to be critical for cancer cell survival during stress stimuli and has been implicated in the resistance to cancer therapeutics [103]. The authors showed that the loss of ISR increases paclitaxel-mediated cell death and that eIF2A is one of the key and essential factors for cancer cell survival under the conditions of paclitaxel-mediated ISR [103]. Knockdown of eIF2A substantially impaired cancer cell survival under the ISR [103]. The authors also found that elevated levels of eIF2A mRNA in patients with breast cancer are correlated with poor prognosis [103].
Sonobe et al. in addition showed that eIF2A aids translation of the C9ORF72 mRNA during the ISR stress in neurons and, specifically, helps translation of the expansion of a hexanucleotide repeat (HRE), GGGGCC motifs (encoding poly(Gly-Ala) repeats in the C9ORF72 gene) [104]. Expansion of HREs, in the C9ORF72 gene is recognized as the most common cause of familial amyotrophic lateral sclerosis (FALS) and frontotemporal dementia (FTD) [104]. It was suggested that translation products of C9ORF72 involving HREs are toxic, may induce ISR and play a critical role in disease pathogenesis [104]. The upstream start codon preceding the repeats that directs the synthesis of poly(Gly-Ala) was identified as CUG [104]. Knockout of eIF2A dramatically impaired the synthesis from the C9ORF71 expanded repeat (GGGGCC) during ISR in neuronal cells [104], therefore suggesting a critical role of eIF2A in repeat associated non-AUG translation disorders [104].
line 437: characterization of a mammalian eIF2A
Response: The text has been changed as suggested by the reviewer.
line 440: The eIF2A knockout mice
Response: The text has been changed as suggested by the reviewer.
line 442: in key initiation pathways and vital for organismalfunctions
Response: The text has been changed as suggested by the reviewer.
line 456: yeast to humans is not a wide phylogenetic spread
Response: The text has been changed as suggested by the reviewer.
line 482: was suggested to drive the protein’sinteractionwitheIF5B
Response: The text has been changed as suggested by the reviewer.
line 510: as well as it abrogating ed eIF2A-mediated
Response: The text has been changed as suggested by the reviewer.
line 513: since it was suggested that the deletion of this region could have also affected eIF2A function
Response: The text has been changed as suggested by the reviewer.
lines 618-620: very confusing sentence
Response: The text was revised and now reads as follows (Lines 638-644):
A largely unexplored area of the research, which is devoted to analysis of the eIF2A function in cellular physiology, integrated stress response and human disease, has recently attracted substantial attention. It is becoming increasingly clear that eIF2A preferentially participates in translation of a specific subset of mRNAs, most likely involving upstream non-canonical initiation codons (uCUG, uUUG) and is supporting the translation of these mRNAs under conditions of ISR (eIF2α phosphorylation) (Figure 5).
lines 626-629: will allow many important questions one to be addressed related to eIF2A function in living systems
Response: The text has been changed as suggested by the reviewer.
Reviewer 2 Report
In their review, the authors focused on the roles of EIF2A in the protein synthesis regulation, mainly at initiation phase. This review is interesting and is of potential importance to the research field. However, I suggest to insert in the review a summarizing table, describing the roles of EIF2A in the alternative mechanisms of protein synthesis regulation, identified to date, with the appropriate references. In addition, the authors could insert some cartoons describing the most representative roles of EIF2A in the alternative mechanisms of protein synthesis regulation.
Author Response
Referee #2 major points:
Referee #2: “In their review, the authors focused on the roles of EIF2A in the protein synthesis regulation, mainly at initiation phase. This review is interesting and is of potential importance to the research field. However, I suggest to insert in the review a summarizing table, describing the roles of EIF2A in the alternative mechanisms of protein synthesis regulation, identified to date, with the appropriate references. In addition, the authors could insert some cartoons describing the most representative roles of EIF2A in the alternative mechanisms of protein synthesis regulation.”
Response: To address the reviewer’s concerns and suggestions, we have now added a new Table 4, summarizing eIF2A participation in various initiation pathways and a new Figure 5 (cartoon, describing the eIF2A pathway).
Round 2
Reviewer 2 Report
I suggest to accept this review in the present form.